

# Validating HY-2A CMR Precipitable Water Vapor Using Ground-based and Shipborne GNSS Observations

Zhilu Wu[1], Yanxiong Liu[1], Yang Liu[1], Jungang Wang[3*,4], Xiufeng He[2], Wenxue Xu[1], Maorong Ge[3,4], Harald Schuh[3,4]

[1]First Institute of Oceanography, Ministry of Natural Resources, Qingdao 266061, China
[2]School of Earth Sciences and Engineering, Hohai University, Nan Jing 211100, China
[3]Department of Geodesy, GeoForschungZentrum, Telegrafenberg, Potsdam 14473, Germany
[4]Institute of Geodesy and Geoinformation Science, Technische Universitat Berlin, Berlin 10623, Germany

*Correspondence to*: Jungang Wang (jgwang@gfz-potsdam.de)

**Abstract.** The calibration microwave radiometer (CMR) onboard Haiyang-2A satellite provides wet tropospheric delays correction for altimetry data, which can also contribute to the understanding of climate system and weather processes. Ground-based Global Navigation Satellite Systems (GNSS) provide precise PWV with high temporal resolution and could be used for calibration and monitoring of the CMR data, and shipborne GNSS provides accurate PWV over open oceans, which can be directly compared with uncontaminated CMR data. In this study, the HY-2A CMR water vapor product is

validated using ground-based GNSS observations of 100 IGS stations along the coastline and 56-day shipborne GNSS observations over the Indian Ocean. The processing strategy for GNSS data and CMR data is discussed in detail. Special efforts were made to the quality control and reconstruction of contaminated CMR data. The validation result shows that HY-2A CMR PWV agrees well with ground-based GNSS PWV with 2.67 mm in RMS within 100 km. Geographically, the RMS is 1.12 mm in the polar region and 2.78 mm elsewhere. The PWV agreement between HY-2A and shipborne GNSS shows a

significant correlation with the distance between the ship and the satellite footprint, with an RMS of 1.57 mm for the distance threshold of 100 km. Ground-based GNSS and shipborne GNSS agree with HY-2A CMR well with no obvious system error.

## 1 Introduction

Sea surface height measurement is mainly implemented by satellite altimetry, where the precise tropospheric delay is

required to correct the atmosphere propagation error in the measured distance between satellite and sea surface. Since the wet delay which can also be quantified by precipitable water vapor (PWV) in meteorology changes rather fast in space and time, the wet delay is measured with onboard water vapor radiometers. On the other hand, PWV is an essential factor in weather and climate system (Randall et al., 2007), especially PWV over oceans plays a paramount role, as more than 80% of atmospheric water vapor comes from the marine. Until now, PWV over ocean is mainly obtained by satellite-borne infrared





sensors (e.g., MODIS, FY-2C) and microwave radiometers (e.g., SSM/I, TMI) (Nelson et al., 2016), and its spatial and temporal resolution can be improved if PWV of altimeter satellites, such as Haiyang-2A (HY-2A) can be included.

HY-2A is a Chinese ocean observation satellite launched on August 15, 2011, operating in a sun-synchronous orbit. The objective of HY-2A is to monitor the dynamic ocean environment, with microwave sensors to detect sea surface wind field, sea surface height, and sea surface temperature. It is equipped with a dual-frequency (13.58 GHz and 5.2 GHz) altimeter, a

calibration microwave radiometer (CMR), a microwave scatterometer, and a scanning microwave radiometer (Jiang et al., 2012). The HY-2A CMR is a nadir-looking passive radiometer with three frequencies (18.7 GHz, 23.8 GHz, and 37 GHz) and the footprint of CMR is ~40 km (Wang et al., 2014). The wet delay and PWV retrieved from CMR measured brightness temperature (TB) were used for satellite altimetry observations correction and marine weather observation as well (Zhang et al., 2015).

Although HY-2A CMR was calibrated in the laboratory before launching (Zhang et al., 2014), due to the quite different space environment, in-orbit validation was carried out subsequently. The onboard validation was mainly conducted by comparing the PWV of HY-2A CMR with that of other altimetry satellites, e.g., Jason 1/2 (Zheng et al., 2014a), or using numerical weather model (NWM) and radiosondes (Wang et al., 2014;Zhao et al., 2016). Furthermore, microwave components will inevitably have aging phenomena, for instance, the Jason-1 and ENVISAT wet tropospheric delay have a

drift of 1 mm per year (Brown, 2013;Obligis et al., 2006). The HY-2A altimetry values also show systematic biases in space and time (Peng 2015; Yang et al. 2016), and the wet tropospheric delay drift is confirmed as one of the reasons (Peng 2015). Moreover, a hardware problem of 18.7 GHz band since June 2017 was reported (Wu et al., 2019). Therefore, the long-term validation and calibration of CMR data are vital for HY-2A applications.

Furthermore, in the observation of HY-2A CMR, only TB measurements over the ocean can be converted into PWV, the

measurement in transition areas, where it happens very often that one footprint covers partly on the sea and partly on the land surface, will be contaminated. Special handling and specific quality control measures should be imposed (Brown, 2010;Fernandes et al., 2003;Fernandes et al., 2010).

GNSS observations have been used for atmosphere sounding since the 1990s (Bevis et al., 1992;Bevis et al., 1994;Manandhar et al., 2018). PWV of an accuracy of 2 mm can be retrieved from ground-based GNSS observations (Gendt

et al., 2004;Li et al., 2015;Wang and Liu, 2019) and has been successfully used for NWM assimilation (Gutman et al., 2004). Meanwhile, GNSS PWV retrieval using moving platforms over the ocean, such as ship and buoy, has been demonstrated with an accuracy of 1-3 mm (Kealy et al., 2012;Rocken et al., 2005;Wang et al., 2019). Therefore, GNSS PWV from coastal stations and especially that from moving platforms over the ocean could be a resource with higher accuracy and resolution for validating and potential calibrating HY-2A CMR data. Liu et al. (2019) investigated the agreement of  shipborne GNSS

PWV over the Indian ocean and HY-2A CMR PWV, where more attention was paid to the GNSS-PWV uncertainty, i.e., the influence of ZTD estimates from different software, the potential error induced by weighted mean temperature and atmospheric pressure, etc. And a strict criteria was applied when choosing the crossovers to ensure the best agreement, which results in only 4 crossovers used. In this paper, we focus more on the HY-2A PWV evaluation on a global scale using



100 ground GNSS stations, and the potential agreement between shipborne GNSS PWV and HY-2A observation under different distance criteria was also discussed.

The paper is arranged as follows. Section 2 presents the data processing strategy, including the PWV retrieval from both ground-based and shipborne GNSS observations, and the reconstruction and quality control of HY-2A PWV of coastal footprints. Section 3 introduces the data used in the study including HY-2A CMR PWV, ground-based and shipborne GNSS observations. Section 4 summarizes the major achievements and conclusions.

## 2 Processing method

In this section, we first introduced the processing strategy of PWV retrieval from ground-based and shipborne GNSS, and the height correction for PWV of ground-based GNSS was also discussed. Then we presented the HY-2A CMR retrieval method and the reconstruction algorithm of coastal HY-2A CMR contaminated data based on European Centre for Medium-Range Weather Forecasts (ECMWF).

### 2.1 GNSS data processing

The ground-based and shipborne GNSS (GPS+GLONASS or GPS) data were processed using the Position And Navigation Data Analyst (PANDA) (Liu and Ge, 2003;Shi et al., 2008) in static and kinematic Precise Point Positioning (PPP) mode, respectively.

GNSS data from 100 IGS stations during DOY 91-147 in 2014 were collected. In the processing, ionosphere-free pseudo-range and phase observations were used with a cut-off elevation angle of 7°. The precise satellite orbit and clock products from GeoForschungsZentrum (GFZ) were used. Satellite and station antenna phase center offsets and variations were corrected using the IGS antenna file (igs08.atx), and the phase wind-up was fixed (Wu et al., 1993). The station displacements caused by solid Earth tides, ocean tides, and pole tide were removed following the IERS 2010 Convention (Petit and Luzum, 2010).

The pressure and temperature from global pressure and temperature (GPT) (Böhm et al., 2007) were used to derive a priori hydrostatic and wet zenith delays with Saastamoinen model (Saastamoinen, 1972). Global mapping function (GMF) (Böhm et al., 2006) was used to map zenith delay to the satellite signal transmitting path. ZTD was estimated as a random walk process with a noise power density of 5 mm / (Kouba and Héroux, 2001), and the horizontal tropospheric gradients every two hours as random walk parameters with 1 cm/ power density.

For the 30-sec resampled shipborne GPS+GLONASS observations, the processing strategy is similar but in kinematic mode, where the receiver antenna coordinates were estimated as epoch-wise parameters. The pressure data was from the shipboard equipment.





Afterward, ZWD was derived by subtracting ZHD from the estimated ZTD where ZHD was calculated from the ERA-Interim layer pressure data provided by ECMWF (Dee et al., 2011). Then ZWD was converted to PWV with the following

equation:

$$PWV = \frac{10^6}{\rho_w R_v [(k_3 / T_m) + k_2']}(ZTD\text{-}ZHD)$$

(1)

where  is the liquid water density;  = 461.495 J/(kg•K), which is water vapor specific gas constant,  =22.97 K/hPa, =375463 K²/hPa (Böhm and Schuh, 2013).  Tm is the atmosphere weighted mean temperature derived from the ERA-Interim product (Davis et al., 1985). The uncertainty of ground-based GNSS PWV is less than 1 mm (Ning et al., 2016) and the uncertainty

of shipborne GNSS PWV is less than 3 mm (Liu et al., 2019).

For comparison, GNSS derived or PWV must be aligned to the same elevation as the HY-2A PWV observations. The orthometric height of stations were calculated as the ellipsoid height from GNSS positioning minus the geoid undulation from the Earth Gravitational Model 2008 (EGM2008). The height correction based on ERA-Interim layer data is as follows: (1) for each station, calculating the ERA-Interim PWV values at the station elevation and at sea level (2) the PWV difference

between these two elevations is used as the PWV height correction; (3) realigning the GNSS PWV to sea level by adding the ERA-Interim derived PWV height correction to the original GNSS PWV.

Fig. 1 shows the biases between ECMWF PWV and GNSS PWV before and after the height correction on all the GNSS stations along with station height. The apparent height-related bias is reduced significantly, especially for the five stations over 1000 m. It should be mentioned that the PWV difference scatters at each station are also improved because of height

correction. In general, the corresponding PWV RMS is reduced to from 5.01 mm to 2.12 mm.

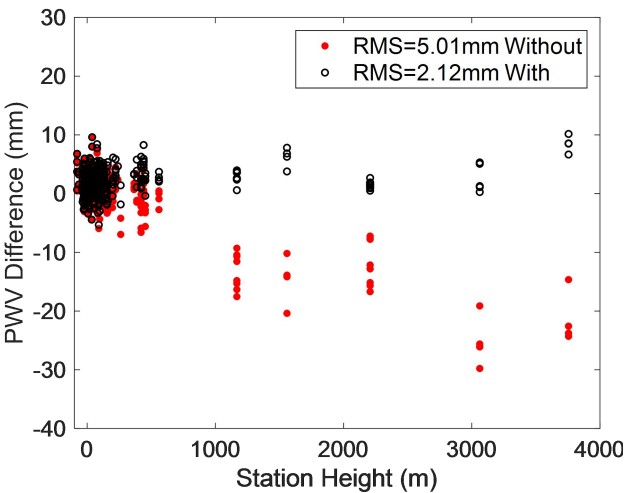

**Figure 1.** PWV biases between GNSS and ECMWF at all the GNSS stations without (red dots) and with (black cycles) height correction.





## 2.2 HY-2A PWV observation processing

The processing of HY-2A CMR contains three steps: antenna temperature calibration, TB adjustment, and PWV retrieval.
The first two steps have been done by the data provider during the conversion from level 0 to level 1 data, which are not
discussed in detail here. In this study, we focus on the validation of level 2 data.

The PWV retrieval algorithm of HY-2A CMR uses an empirical regression model based on the TB values from level 1
and PWV from the global database (Robinson, 2004). There are mainly two regression models to retrieve water vapor
products: neural network algorithm and log-linear regression (Brown et al., 2004;Obligis et al., 2006;Thao et al., 2015). For
HY-2A CMR data, the log-linear regression algorithm is widely used (Wang and Zhang, 2008;Zheng et al., 2014b). The
model reads as:

$$PWV = a_0 + a_{18.7} \cdot \ln(b_{18.7} - TB_{18.7}) + a_{23.8} \cdot \ln(b_{23.8} - TB_{23.8}) + a_{37} \cdot \ln(b_{37} - TB_{37}) \tag{2}$$

where $TB_{18.7}$, $TB_{23.8}$, $TB_{37}$ are the TB in K of the three frequencies (18.7 GHz, 23.8 GHz, and 37 GHz), respectively. $a_0$, $a_{18.7}$,
$b_{18.7}$, $a_{23.8}$, $b_{23.8}$, $a_{37}$, $b_{37}$ are the coefficients in the retrieval models. These coefficients must be estimated using external PWV
datasets, e.g., NWM, radiosonde profiles, or previous satellite altimetry missions. This procedure is referred as the
calibration of CMR data and should be carried out carefully and updated in time in order to obtain accurate PWV
observations. The uncertainty of the CMR PWV dataset is less than 3 mm according to the 7 years in-flight CMR
observations (Wu et al., 2019).

For each transit of HY-2A to a ground GNSS station, there are a number of crossover points at different distances to the
station. To avoid the potential large bias caused by a single point, the crossover points within 100 km to the GNSS station
are used with the inverse distance weighting (IDW) interpolation:

$$PWV(s) = \frac{\sum_{i=1}^{N} \omega_i(s) \, PWV_i}{\sum_{i=1}^{N} \omega_i(s)} \tag{3}$$

$$\omega_i(s) = \frac{1}{d(s, s_i)} \tag{4}$$

where $PWV(s)$ is the virtual measurement of HY-2A CMR at the GNSS station, $\omega_i(s)$ is the weight value, $PWV_i$ is the
PWV of HY-2A crossover point. $d(s, s_i)$ is the distance between the crossover point and the GNSS station, which is
always larger than 0 since the GNSS stations locate several kilometers away from the coastline.

From the weight in Eq. (3), we need the CMR observations of the crossover points geometrically close to the ground
station. However, the quality of these observations could be rather poor, as they may contain the contribution of reflected
signals from both land/ice and ocean, which have different emissivity character. Therefore, HY-2A CMR data quality
control is very crucial. First of all, the footprints flagged as "land" and "ice" must be excluded in advance. Then, the





footprints flagged as "ocean" close to coastal regions should be checked carefully, due to the potential land contamination where the 40 km footprint may cover both ocean and land but flagged as "ocean".

The sampling interval of HY-2A CMR is 1s and the moving speed of the footprint is approximately 6 km/s, thus the variation of HY-2A CMR PWV between consecutive epochs should be very smooth. Therefore, a linear fitting of HY-2A

CMR PWV could be used for quality control in principle. However, for the regions with complex terrain, such as archipelago, where there are more outliers than useful crossover points, the linear fit does not work. Therefore, a reliable external dataset is necessary for quality control. In this study, the vertical integral of water vapor (VIWV) from ERA-Interim surface product was used, where the PWV differences between HY-2A CMR and ECMWF at crossover points should be small and stable and those with extremely large values were considered as invalid points.

The reconstruction of contaminated HY-2A PWV is usually implemented in the following two scenarios: (1) near the coastline where clean points are available only on the ocean side; (2) near the peninsula or small islands, where clean points before and after the contaminated point are available. The algorithm for the reconstruction of contaminated HY-2A CMR PWV can be summarized as follows (Fernandes et al., 2003):

$$PWV_{hy\_rec} = PWV_{ecmwf} + f(PWV_{hy\_clean}, PWV_{ecmwf\_clean}) \tag{5}$$

where $PWV_{hy\_rec}$ is the reconstructed HY-2A PWV at the crossover point, and $PWV_{ecmwf}$ is the ECMWF PWV at this crossover point.

In Eq. (5), $f(PWV_{hy\_clean}, PWV_{ecmwf\_clean})$ is a linear function to calculate the PWV differences between HY-2A CMR and ECMWF at the contaminated crossover points based on the differences of all the clean points. For the first case, the difference is extrapolated using the PWV differences of the clean points on one side; while for the second case, it is

interpolated using the PWV differences of clean points on both sides.

The HY-2A CMR observations (DOY 123 and 128, 2014) in Southeast Asia were used as an example to illustrate the coastal PWV reconstruction. The two trajectories of HY-2A traversed Malaysia and Indonesia in parallel, shown in the right panel of Fig. 2, including three representative terrain types, i.e., continental coast, peninsula, and islands. Meanwhile, as the PWV value in this low-latitude coastal area is rather large (>50 mm), a careful reconstruction should be implemented.

The reconstruction of contaminated HY-2A CMR data was carried out with the ECMWF as the background field. As shown in the left panel of Fig. 2, the HY-2A PWV observations at coastal areas could be largely biased up to 100 mm, marked with red dots from the clean observations with green dots. By applying the aforementioned reconstruction algorithm, the reconstructed PWV observations show a much better agreement with the clean observations. The average value of the PWV biases between HY-2A CMR and ECMWF was reduced from 5.52 mm to 2.78 mm, while the standard deviation (STD) was

reduced from 13.18 mm to 2.71 mm.



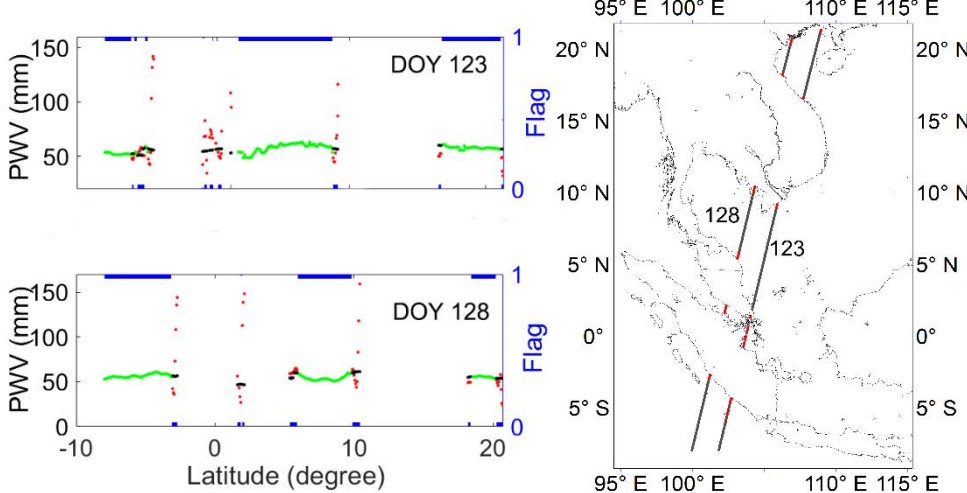

**Figure 2.** HY-2A PWV observations on DOY 123 (left-upper) and DOY 128 (left-bottom) in 2014, where the clean, contaminated, and reconstructed observations are shown as green dots, red dots, and black dots, respectively. The right Y-axis is the label of clean points (value 1) and contaminated points (value 0). The right panel illustrates the satellite trajectory on DOY 123 and DOY 128, where the clean and contaminated footprints are shown in gray and red dots, respectively.

## 3 Dataset

In this study, the HY-2A CMR PWV was compared to the ground-based and shipborne GNSS PWV. The ground-based GNSS data on the period of DOY 091-146, 2014 was collected and used, which is the same as the ship cruise. In this section, the datasets, i.e., the HY-2A CMR PWV observations, the ground-based GNSS observations, and the shipborne GNSS observations, are introduced.

### 3.1 HY-2A CMR PWV observations

The CMR PWV products used were provided by the National Ocean Satellite Application Center (NSOAS), Ministry of Natural Resources (MNR) of China. The PWV were retrieved from the TB observation using Eq. (2). Two months HY-2A CMR in-orbit data for the period of DOY 91-147, 2014 were used. The raw HY-2A PWV observations were processed with a complicated outlier detection method and coastal observation reconstruction was implemented on the HY-2A PWV observations, as mentioned in Section 2.2.

### 3.2 Ground-based GNSS observations

Fig. 3 shows the distribution of 100 IGS stations used for the HY-2A CMR PWV comparison, including 46 stations located on islands and 54 stations located on mainland coastline. Each station has at least one crossover point within 100 km compared with the HY-2A sub-satellite point during the experiment period. Most of the stations are below 200 m, and five





stations are above 1000 m. The GPS observations were used for all these stations, and the GLONASS observations were used whenever available.

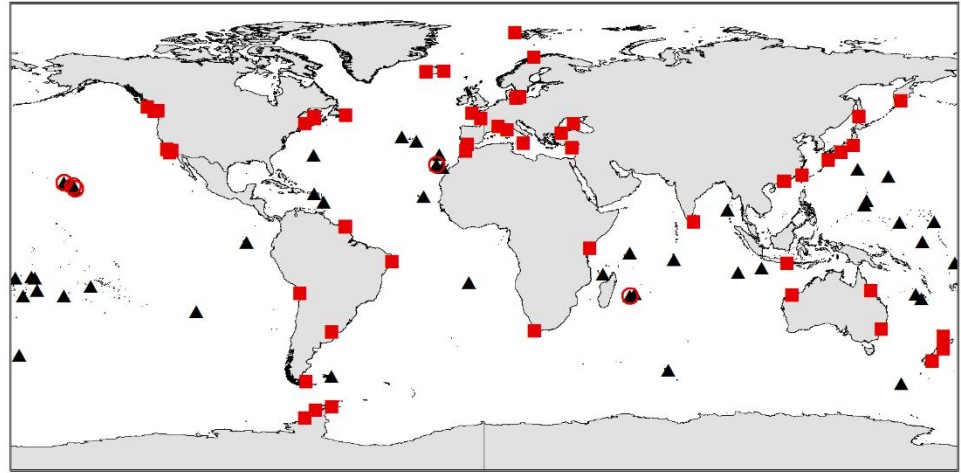

**Figure 3.** The distribution of GNSS sites within 100 km compared to HY-2A sub-satellite points, including 46 on islands (black triangle) and 56 on the mainland coastline (red square). Five of them (red circle) are higher than 1000 m.

### 3.3 Shipborne GNSS observations

The scientific survey of the Indian Ocean during DOY 91-146 in 2014 and the ship trajectory is shown in Fig. 4. The voyage started from Guangzhou, China, and went through the Sunda Strait into the Indian Ocean, sailing along the equatorial and arrived in Sri Lanka, finally returned to Guangzhou via the Malacca Strait. The ship was equipped with a TPS NET-G3A GNSS reference receiver to collect 1 Hz GPS+GLONASS data, where the choke ring antenna was used to reduce the multipath effect.

HY-2A satellite moves very fast (~6 km/s) while the speed of the ship is low (maximum ~35 km/h) and the ship track lacks regularity, the crossover points between satellite footprint and ship track are scarce in both time and space. To have more crossover points for comparison, the thresholds in distance and time difference were set to 200 km and 2 hours, respectively. Applying the thresholds, finally, 11 crossover events with 629 crossover points were found, which are shown with red dots in Fig. 4. The crossing time of each crossover event is presented next to the event mark in green text. With such a large number of crossover points, we can also further analyze the impact of different distance thresholds on the PWV comparison. It should be noted that the discontinuity of ship track was caused by the missing of GNSS observations.

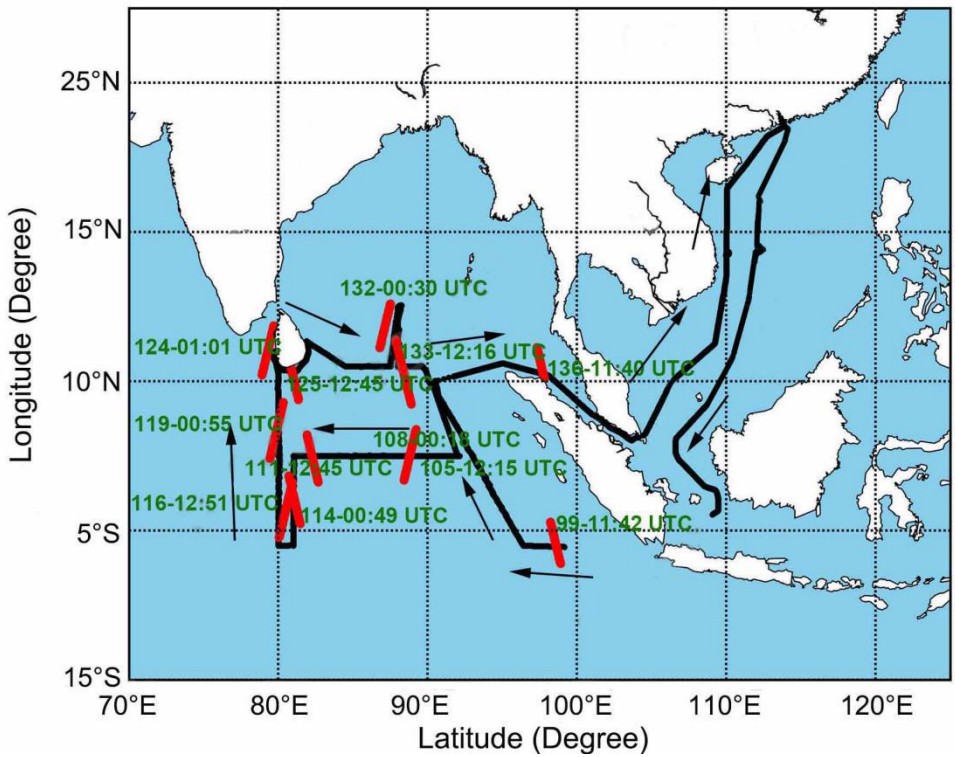

**Figure 4.** The ship trajectory (black line) and HY-2A crossover points (red dots). There are 11 crossover events and the averaged overlapping time of each event is marked in green text.

## 4 Results and Discussion

In this section, the HY-2A CMR PWV was compared to GNSS PWV and the results were presented, including the comparison to ground-based GNSS PWV and shipborne GNSS PWV. The PWV height correction was applied at all the IGS stations comparison, and the HY-2A PWV observations in coastal regions were reconstructed to avoid the land contamination.

### 4.1 HY-2A PWV validation using ground-based GNSS

The HY-2A PWV observations of two months data were compared with the GNSS PWV on the 100 coastal stations. Analysis of the comparison results of the two sets is in the top panel of Fig. 5, while the detailed statistics of all crossover points in different latitude regions are shown in the bottom panel, where the PWV differences in polar regions (>66.5°), tropical area (23.5°N - 23.5°S) and mid-latitude regions (23.5° - 66.5°) were presented.





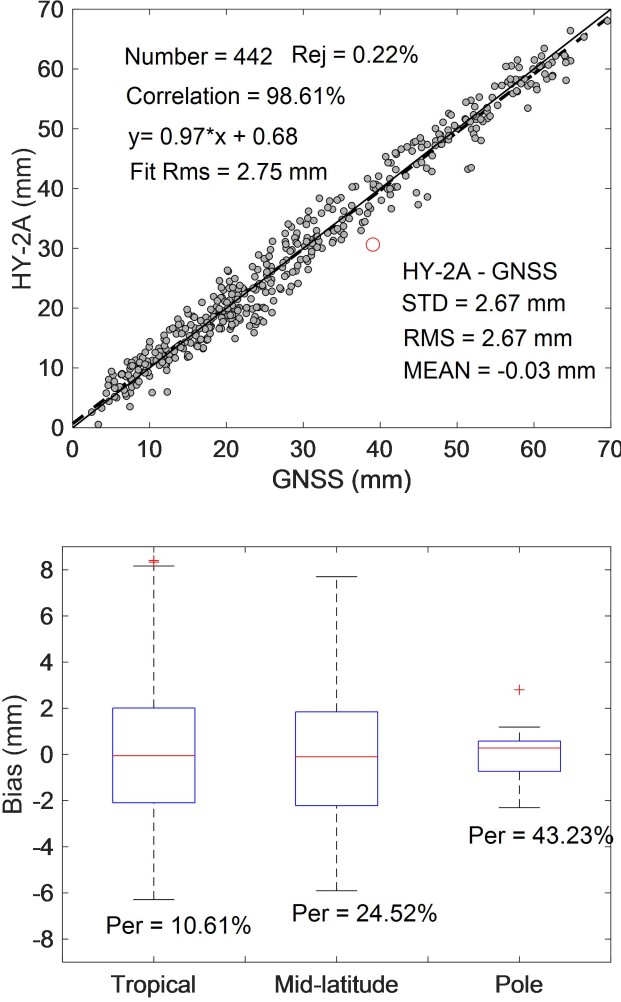

**Figure 5.** PWV difference between HY-2A and ground GNSS. The top panel shows PWV scatter diagram of HY-2A observations and GNSS results, the linear fit result is the solid line and the dashed line is the reference line, and red circles are outliers. The bottom panel shows boxplot of the difference between HY-2A and GNSS at tropical regions, mid-latitude regions, and pole regions. The blue box describes the upper and lower quartiles, the red line inside the box is median values, the whiskers are 1.5 times the interquartile range (IQR), and the red open cross-hatches describe data outliers, Per means the average ratio of the RMS to derived PWV of the region.

The top panel of Fig. 5 shows the scatter of HY-2A CMR PWV and ground-based GNSS PWV and linear fit result, in which about 0.22% of the total points with a difference larger than three times of STD were excluded. The HY-2A PWV shows a good agreement with the GNSS PWV with an average bias of -0.03 mm, and the RMS is 2.67 mm. No systematic bias is revealed and a high correlation coefficient of 98.61% is achieved. As shown in the bottom panel, the difference in polar regions is smaller than that in other regions; the upper and down quartile are -2.31 mm and 1.18 mm, respectively, with

an average value of 0.27 mm and an RMS value of 1.12mm. The relative PWV error (PWV bias / PWV value) in polar region (43.23%) is much larger than that in other regions. On the other hand, the PWV RMS in lower and middle latitude regions is 2.78 mm, partially because of the large PWV content in these regions. It should also be noted that all the stations used in this comparison are located in coastal regions, which usually has a larger PWV. The PWV agreement between HY-2A and ground-based GNSS does not show an obvious correlation with latitude.

Moreover, for each GNSS station, the statistics of PWV differences of all the crossover points at different times were calculated and together with the average distance to the crossover points. The relationship between the PWV differences and the averaged distance and GNSS station height is shown in Fig. 6. As shown in the left panel of Fig. 6, both the average value and STD of the PWV differences between HY-2A and GNSS show no correlation with the averaged distance ranging from 40 km to 95 km, indicating that the HY-2A reconstructed PWV observations are free of land contamination. The right

panel confirms that the PWV differences have no correlation with station height, which means that the PWV height correction at ground-based GNSS stations is effective.

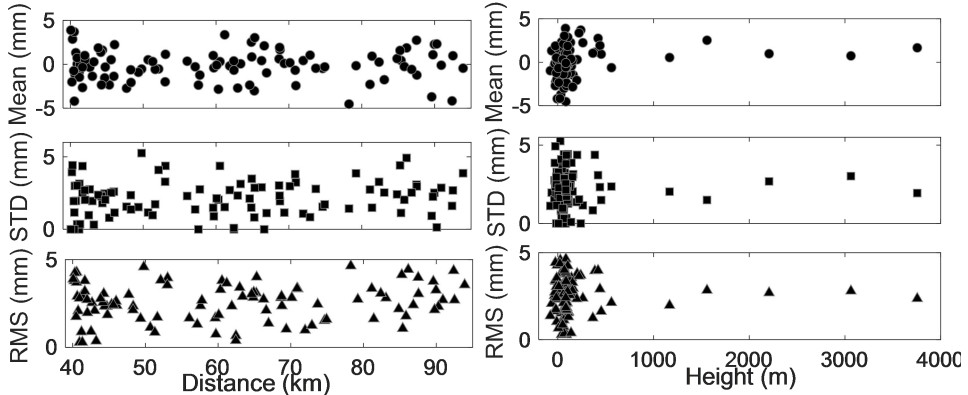

**Figure 6.** The statistics of PWV differences between HY-2A and ground-based GNSS w.r.t. the distance between HY-2A footprint (left panel) and GNSS stations and the station height (right panel).

**4.1 HY-2A PWV validation using shipborne GNSS**

Although ground-based GNSS can provide long-term PWV with certain accuracy for the validation of HY-2A CMR, the validation could still suffer from contaminated CMR data by the signal on the land, long distance between GNSS station and footprint, and GNSS height, especially for data of higher precision. The comparison result could be affected by the residual error after correction. On the other hand, shipborne GNSS observations in open-sea regions provide an accurate and direct

method for the satellite altimetry comparison and validation. The validation of HY-2A PWV using shipborne GNSS observations was presented in this section.

As mentioned before, with the threshold of 200 km and 2 hours, the number of the crossover events between HY-2A and GNSS are 11 days with 629 crossover points in total, which is shown in Fig. 4. For each crossover event, the PWV




observations larger than three times of the STD value of the differences were removed as outliers, i.e., the 3σ criterion.
Among the 629 crossover points, ~4.8% were removed as outliers and the useful number is 599.

To investigate the impact of the space threshold on the PWV validation, the PWV differences of the crossover points defined with space threshold of 200 km, 150 km, 100 km, and 50 km are presented in Fig. 7. As shown in Fig. 7, the PWV agreement between HY-2A CMR and shipborne GNSS decreases with the increase of the corresponding distance threshold. The RMS within 200 km, 150 km, 100 km, and 50 km are 2.89 mm, 1.78 mm, 1.53 mm, and 0.89 mm, respectively, and the
correlation increases from 77% to 98.2%. The outliers (red dots) decrease from 4.77% to 0% when the distance threshold is getting smaller. The linear fit also shows a better agreement when the distance is shorter.

The average bias is 0.32 mm, meaning that there is no obvious systematic bias between HY-2A PWV and shipborne GNSS PWV. Since the variation of PWV is relatively slow over the ocean, the average bias remains small even though the distance comes to 200 km. The agreement between shipborne GNSS and HY-2A data is better than that of the ground GNSS
result, which could be caused by the potential residual error of the ground-based GNSS stations due to the PWV height correction and that of the HY-2A observations due to the data reconstruction; the complex topography in coastal regions could be another reason. Overall, the PWV differences of HY-2A CMR data concerning ground and shipborne GNSS is 2.67 mm and 1.53 mm in RMS for the distance threshold of 100 km.

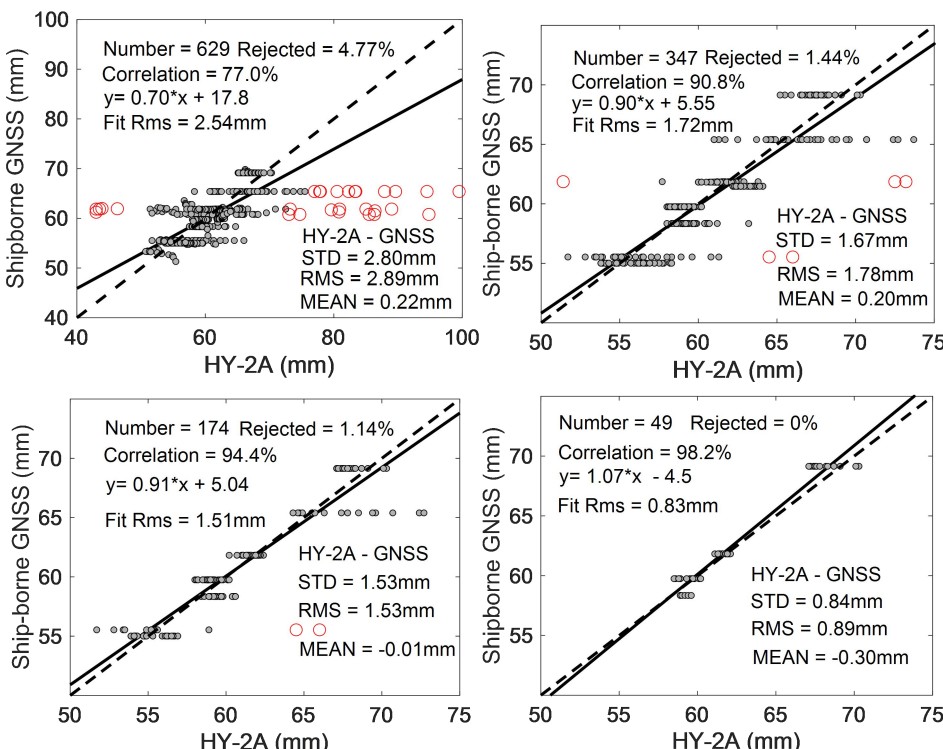

**Figure 7.** PWV comparison between HY-2A CMR and shipborne GNSS for crossover points with a distance threshold of 200 km (upper-left), 150 km (upper-right), 100 km (bottom-left), and 50 km (bottom-right). The red circles are for outliers, and the linear fit result is





presented as a solid line with its reference in dash line. For each panel, the linear fit is shown in upper-left and the comparison statistics are shown in lower-right.

## 5 Conclusion

Water vapor over oceans is essential for both the altimeter correction and the understanding of climate system and weather processes. Therefore, retrieving and validating HY-2A CMR WPV become critical.

In this study, we focus on the validation of PWV from HY-2A CMR using GNSS PWV, including 100 ground GNSS stations and a 56-day shipborne GNSS observation campaign in 2014. The HY-2A PWV observations in coastal regions were carefully checked and those suffer from land contamination were reconstructed using PWV products from NWM. The

PWV height correction was applied to the ground-based GNSS stations to remove the height-related variations. The result shows that HY-2A PWV agrees well with the ground-based GNSS PWV with an average bias of -0.03 mm and an RMS value of 2.67 mm. The PWV difference in polar regions is smaller than that in the other areas due to the low PWV content in the polar areas. Comparison with the shipborne GNSS PWV, HY-2A PWV shows an agreement of 0.89 mm in RMS for the 49 crossover points within 50 km, and an agreement of 2.89 mm in RMS for the 629 crossover points within 200 km. Both

ground-based GNSS PWV and shipborne GNSS PWV show a good agreement with the HY-2A PWV observations without any obvious systematic bias.

Based on the validation result, GNSS PWV, especially that retrieved from shipborne data over open oceans, can play a critical role in the calibration of HY-2A CMR data. Since HY-2A, after its operation for more than seven years, is facing the problem of inaccurate CMR data, e.g., biased PWV and ZWD caused by the aging of observation device. The new

calibration using GNSS PWV can provide precise HY-2A CMR data for both altimeter correction and meteorological study.

*Acknowledgments* We would like to thank NSOAS for providing HY-2A data, South China Sea Institute of Oceanology for the research vessel cruise, ECMWF for NWM data, IGS for GNSS observations and GFZ for precise orbit products.

*Data availability.* The source codes for the validation and the data of HY-2A and shipborne GNSS used in this study can be provided by the corresponding authors upon reasonable request.

*Author contributions.* ZW and JW co-designed this research; ZW, JW, XH, YL and XW conducted the data analyses; ZW wrote the manuscript. All authors discussed the experimental results and helped revise the manuscript.


*Competing interests.* The authors declare that they have no conflict of interest.



*Financial Support* This research is supported by the National Natural Science Foundation of China (41876106), Basic Scientific Fund for National Public Research Institutes of China (2018Q04) and the National Key R&D Program of China (2017YFC1405100). Mr. Jungang Wang is financially supported by China Scholarship Council (CSC. File No. 201606260035).

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
