# Peer review of "Validating HY-2A CMR Precipitable Water Vapor Using Ground-based and Shipborne GNSS Observations"

_Atmospheric Measurement Techniques, 2019_

## Referee Comment (RC1) · Anonymous Referee #1 · 13 Apr 2020

The paper "Validating HY-2A CMR Precipitable Water Vapor Using Ground-based and Shipborne GNSS Observations" contain comparative analysis between GNSS PWV and PWV obtained from HY-2A calibration microwave radiometer, which may be used for the purpose of validating satellite data. It also include procedure for reconstruction of coastal PWV, which quality suffer due to the contribution of various land and water emissivity. These both aspects are important for reliable use of spaceborne equipment for PWV measuring. Although paper is structured in proper way it causes problems in the correct and transparent understanding of its content. In my opinion this results from a heterogeneous description of the results. In addition, the Authors quite unsystematically approach the selection of results worthy of additional discussion or comment in the text. In consequence, it seems that all obtained results are in very good agreement, which in my opinion is not so obvious. This is surprisingly, since detailed analysis would probably bring some interesting results, which could be include to the conclusion section.

Detailed issues:

108-110 and Fig1. I am not sure if there is a point to include this information. Height correction is simply necessary due to the characteristic of vertical distribution of water vapor and it is obvious that higher differences in elevation results in higher differences in PWV.

Line 252-253 "validation could still suffer from contaminated CMR data by the signal on the land, long distance between GNSS station and footprint, and GNSS height, especially for data of higher precision" – I am quite confused here. Several lines above it is show (on Figure 6) and pointed in the text, that distance up to 90 km has no significant impact on the differences, as well as station height due to the proper correction conducted at the beginning of calculations. Therefore I am not sure what author would like to say here. What kind of "long distance" is it (more than 90 km?). What GNSS elevation is mentioned here as a source of error, since as it was written in line 244-246 "The right panel confirms that the PWV differences have no correlation with station height, which means that the PWV height correction at ground-based GNSS stations is effective". Maybe it would be more clear if the Author would specify this "data of higher precision". Without specifying these information this paragraph is in contradiction with what the Authors have wrote before.

line 264 - "The RMS within 200 km, 150 km, 100 km, and 50 km are 2.89 mm, 1.78 mm, 1.53 mm, and 0.89 mm, respectively". Here Authors underline that the distance has impact on the differences between HY-2A and shipborne GNSS. This is in contrast to what they have wrote in line 242 – 243 "both the average value and STD of the PWV differences between HY-2A and GNSS show no correlation with the averaged distance ranging" (the mean RMS for ground-bases GNSS was 2.67 mm). Of course

at this point we have larger distance (up to 200 km), but the RMS for distance 100 km is for about 70% higher than for distance 50 km, while for similar distance during comparison HY-2A to land GNSS (45 km to 90 km) it was 'no correlation'. There is also no comment about differences obtained for ground-based and shipborne GNSS. The mean RMS for shipborne would be about 1.4 mm (Authors did not provide this value), which is two times smaller than the mean RMS for ground base GNSS. In my opinion this indicate that procedure for PWV coastal reconstruction is not without errors. There is no 'ideal' way to reconstruct valid data, but this should be clearly pointed by the Authors. I would appreciate if Authors could provide some explanation about this.

Line 267 " The average bias is 0.32 mm, meaning that there is no obvious systematic bias between HY-2A PWV and shipborne GNSS PWV." In case of ground-base GNSS the mean bias was -0.03 mm. Since this value (-0.03mm) was expressed as 'good agreement', and 0.32 is also expressed as 'no obvious bias' where according to the Authors is a threshold, after which we can talk about systematic bias? In addition, the information about RMS w.r.t. distance were provide – why there is no information about bias w.r.t. distance?

According to Figure 7, mean difference between HY-2A and GNSS is 0.22 mm, 0.20 mm, -0.01 mm and -0.30mm, for threshold distance equal to 200 km, 150 km, 100 km an 50 km. Since 'mean differences' are simply biases, where the value of mean bias equal to 0.32 come from? In addition from what Authors mention results that the biases are not obvious and rather indicate high compliance between PWV from shipborne GNSS and from HY-2A. In my opinion the bias is clearly positive and clearly negative between the two extreme thresholds (50 km and 200 km). I do not see any comment about that. This is strongly related to the Authors 'threshold' for significance bias, which I mention before.

Generally, the results for ground-based and shipborne GNSS should be rewritten to avoid such misunderstandings as I mentioned above.

Section 5 is not conclusions section. Is rather a (very) short summary of obtained results, without critical and interesting findings which are necessary in such section. There is also no references to similar studies.

After all, could Authors provide more explanation about including shipborne GNSS in this paper. It is not clear why they decided to analyze this data, since (from the selection of only coastal ground-based GNSS stations) the main activity of this study is rather related to the 'problematic' coastal area, than to the clear oceans. Please add some information to the text.

technical corrections:

Figure 1, Please avoid in legend such shortcuts as "With" and "Without". The Figure should be prepare in the way, which will make it possible for anyone to understand it content, without referring to the text

Figure 4 has to be improved. The crossover time cannot overlap the crossover point, especially when green and red colors are used, because it makes it difficult to read them. The crossover time should not also overlap with the ship trajectory.

---

## Author Comment (AC1) · 11 May 2020

Dear reviewer,

Thank you for the constructive and encouraging comments regarding our manuscript. We have enclosed a carefully revised manuscript according to the comments and suggestions provided. We also provide an item-by-item response to all comments.

Yours Sincerely, Zhilu Wu

Response to Reviewer

108-110 and Fig1. I am not sure if there is a point to include this information. Height correction is simply necessary due to the characteristic of vertical distribution of water

vapor and it is obvious that higher differences in elevation results in higher differences in PWV. Response: Thank you for the suggestion. The height correction is necessary for GNSS PWV because of the vertical distribution of water vapor. The correction method we used may affect further validation, therefore it is necessary to briefly present the efficiency/precision of this method. The information from Figure 1 is needed.

Line 252-253 "validation could still suffer from contaminated CMR data by the signal on the land, long distance between GNSS station and footprint, and GNSS height, especially for data of higher precision" – I am quite confused here. Several lines above it is show (on Figure 6) and pointed in the text, that distance up to 90 km has no significant impact on the differences, as well as station height due to the proper correction conducted at the beginning of calculations. Therefore I am not sure what author would like to say here. What kind of "long 9distance" is it (more than 90 km?). What GNSS elevation is mentioned here as a source of error, since as it was written in line 244-246 "The right panel confirms that the PWV differences have no correlation with station height, which means that the PWV height correction at ground-based GNSS stations is effective". Maybe it would be more clear if the Author would specify this "data of higher precision". Without specifying these information this paragraph is in contradiction with what the Authors have wrote before. Response: Thank you for the suggestion. In the line 252-253"validation could still suffer from contaminated CMR data by the signal on the land, long distance between GNSS station and footprint, and GNSS height, especially for data of higher precision", the Figure 6 shows the land contamination of satellite-borne PWV is corrected properly, and the reconstruction method used in the paper shows no relation with the distance. In this figure, we show no significant differences with the distance varying from 40 km to 90 km, which does not mean that the differences are comparable to that of co-located points (e.g., less than 5 km ). The ground-based GNSS sites are still far away from the ocean with tens of kilometers (minimum distance of 40 km), therefore the distance between sub-satellite points and GNSS sites still exists. Since the characteristic of the horizontal distribution of water vapor, this part of error can affect the comparison. On the other hand, Figure

6 shows that the PWV differences between shipborne GNSS and HY-2A increase significantly (from 0.89 mm to 1.53 mm) with the distance criteria increases from 50 km to 100 km. As there is no land containment in the shipborne case, it indicates that the PWV agreement is sensitive to the horizontal distance. In the case of ground-based GNSS, however, we do not have so many observations within 50 km to the HY-2A footprint, and we have to consider the coupled effects of potential land containment, distance-induces bias, the model errors of height correction and reconstruction. We have revised the line of 244: ". . . are free of land contamination. However, it should be noted that in the left panel the distance is varying from 40 km to 90 km, thus this conclusion does not indicate the case of distance less than 40 km. The right panel . . ." Also, the vertical distribution of water vapor affect the comparison, the height correction is processed before comparing ground-based GNSS PWV and satellite-borne PWV. We revised "especially for data of higher precision" to "especially for higher accuracy of validation based on GNSS".

line 264 - "The RMS within 200 km, 150 km, 100 km, and 50 km are 2.89 mm, 1.78 mm, 1.53 mm, and 0.89 mm, respectively". Here Authors underline that the distance has an impact on the differences between HY-2A and shipborne GNSS. This is in contrast to what they have written in line 242 – 243 "both the average value and STD of the PWV differences between HY-2A and GNSS show no correlation with the averaged distance ranging" (the mean RMS for ground-based GNSS was 2.67 mm). Of course, at this point, we have larger distance (up to 200 km), but the RMS for distance 100 km is for about 70% higher than for distance 50 km, while for similar distance during comparison HY-2A to land GNSS (45 km to 90 km) it was 'no correlation'. There is also no comment about differences obtained for ground-based and shipborne GNSS. The mean RMS for shipborne would be about 1.4 mm (Authors did not provide this value), which is two times smaller than the mean RMS for ground base GNSS. In my opinion, this indicates that procedure for PWV coastal reconstruction is not without errors. There is no 'ideal' way to reconstruct valid data, but this should be clearly pointed by the Authors. I would appreciate if the Authors could provide some explanation about this. Response: Many

thanks for the advice. In line 264 -"The RMS within 200 km, 150 km, 100 km, and 50 km are 2.89 mm, 1.78 mm, 1.53 mm, and 0.89 mm, respectively", we discussed the agreement between shipborne GNSS and HY-2A in different distance. While in line 242–243"both the average value and STD of the PWV differences between HY-2A and GNSS show no correlation with the averaged distance ranging", we discussed the mean value and STD of difference between ground-based GNSS and HY-2A after the reconstruction, which mainly focuses on the evaluation of the reconstruction method. When ground-based GNSS sites and sub-satellite points are closer, the HY-2A data are more easily be contaminated. Therefore, when the distance is closer, the agreement between ground-based GNSS and HY-2A might be getting worse. Figure 6 shows no correlation between the agreement and the distance (varying from 40 km to 90 km), which means the reconstruction method is effective in this distance region. Indeed there is no 'ideal' method to reconstruct valid data. Thanks for your advice. We have clearly pointed out this in the revised manuscript: "It should be pointed out that there is no 'ideal' method to reconstruct the valid PWV data in the coastal region, but it is still necessary to spare no efforts to investigate any useful method to derive 'clean' data for inter-technique comparison and validation"

Line 267 " The average bias is 0.32 mm, meaning that there is no obvious systematic bias between HY-2A PWV and shipborne GNSS PWV." In the case of ground-base GNSS the mean bias was -0.03 mm. Since this value (-0.03mm) was expressed as 'good agreement', and 0.32 is also expressed as 'no obvious bias' where according to the Authors is a threshold, after which we can talk about systematic bias? In addition, the information about RMS w.r.t. the distance were provide – why there is no information about bias w.r.t. distance? Response: Thanks a lot for your comments and we are truly sorry for the misleading interpretation. The 0.22 mm PWV bias between HY-2A and shipborne GNSS is indeed much larger the -0.03 mm bias between ground-based GNSS and HY-2A. We have added new information about the bias w.r.t. the distance in the revised manuscript after line 267: "When the distance is getting closer (from 200 km to 100 km), the mean bias between shipborne GNSS and HY-2A is getting closer

to zero (varies from 0.22 mm to -0.01 mm). The average bias is -0.22 mm within 50 km, which might be caused by the limited sample number (49 crossovers)."

According to Figure 7, mean difference between HY-2A and GNSS is 0.22 mm, 0.20 mm, -0.01 mm and -0.30mm, for threshold distance equal to 200 km, 150 km, 100 km an 50 km. Since 'mean differences' are simply biases, where the value of mean bias equal to 0.32 come from? In addition from what Authors mention results that the biases are not obvious and rather indicate high compliance between PWV from shipborne GNSS and from HY-2A. In my opinion the bias is clearly positive and clearly negative between the two extreme thresholds (50 km and 200 km). I do not see any comment about that. This is strongly related to the Authors 'threshold' for significance bias, which I mention before. Generally, the results for ground-based and shipborne GNSS should be rewritten to avoid such misunderstandings as I mentioned above. Response: Thank you so much for your advice again, we are truly sorry for the mistake. The mean bias 0.32 mm should be 0.22 mm for the distance of 200 km. In line 267, we revised the manuscript accordingly: "The average bias is 0.22 mm for the distance of 200 km with a much larger STD value (2.80 mm)." The mean bias in different distance changed from positive to negative between the two extreme thresholds (50 km and 200 km), we added the explanation after line 267 as mentioned in the last comment.

Section 5 is not conclusions section. Is rather a (very) short summary of obtained results, without critical and interesting findings which are necessary in such section. There is also no references to similar studies. After all, could Authors provide more explanation about including shipborne GNSS in this paper. It is not clear why they decided to analyze this data, since (from the selection of only coastal ground-based GNSS stations) the main activity of this study is rather related to the 'problematic' coastal area, than to the clear oceans. Please add some information to the text. Response: Thank you so much for the suggestion. We revised the manuscript accordingly: "Water vapor over oceans is essential for both the altimeter correction and the understanding of climate system and weather processes. Therefore, retrieving and validating HY-2A

CMR PWV becomes critical. HY-2A PWV is mainly validated with NWM and other satellites (Wang et al., 2014; Zhao et al., 2016). Liu et al. (2019) investigated the agreement of shipborne GNSS and HY-2A CMR PWV, where more attention was paid to the GNSS-PWV uncertainty. In this study, we focus on the HY-2A PWV evaluation on a global scale validation with GNSS observations. The HY-2A PWV observations in coastal regions were carefully checked and those suffer from land contamination were reconstructed using PWV products from NWM. The PWV height correction was applied to the ground-based GNSS stations to remove the height-related variations. The result shows that HY-2A PWV agrees well with the ground-based GNSS PWV with an RMS value of 2.67 mm with no obvious system bias. Besides, we compared shipborne-GNSS-derived PWV and HY-2A PWV, which shows the difference of 1.53 mm in RMS within 100 km. The shipborne-GNSS reveals a better agreement than the ground-based result, which because the residual error from the HY-2A reconstruction and ground-based GNSS PWV height correction, and the complex topography in coastal regions could be another reason. Since HY-2A, after its operation for more than seven years, is facing the problem of inaccurate CMR data, e.g., biased PWV and ZWD caused by the aging of observation device. Although the agreement between HY-2A and ground-based GNSS is relatively worse, the ground-based could provide long-term observation globally with relatively high accuracy. With the supplement of shipborne GNSS observations, the new validation method using GNSS observation can play a critical role in the calibration of HY-2A CMR data, and improve the accuracy of HY-2A data for both altimeter correction and meteorological study."

We added the explanation of the reason to analyze shipborne GNSS after line 254: "The coastal GNSS can be combined with shipborne kinematic GNSS, which can also obtain high accuracy WTC (Wang et al. 2019), and shipborne GNSS observations in open-sea regions provide an accurate and direct method for the satellite altimetry comparison and validation, which is free of any land contamination or height correction error. The shipborne GNSS observations could be a very good supplement for the validation using GNSS observation and extend the method to open-sea. More than

160,000 vessels are sailing across the ocean daily (https://www.marinetraffic.com), and these data can also be used for calibration if the vessels are equipped with geodetic GNSS receiver and antenna. "

Wang, J., Zhang, J., Fan, C., and Wang, J.: Validation of the "HY-2" altimeter wet tropospheric path delay correction based on radiosonde data, Acta Oceanologica Sinica, 33, 48-53, 2014. Wang J et al. (2019) Retrieving Precipitable Water Vapor From Shipborne Multi-GNSS Observations. Geophysical Research Letters, 46(9):5000–5008. Zhao, J., Zhang, D., Wang, Z., and Li, Y.: The validation of HY-2A ACMR retrieval algorithms and product, Geoscience and Remote Sensing Symposium (IGARSS), 2016 IEEE International, Beijing, China, 2016, 411-413, 2016. Liu, Liu, Y., Chen, G., and Wu, Z.: Evaluation of HY-2A satellite-borne water vapor radiometer with shipborne GPS and GLONASS 355 observations over the Indian Ocean, GPS Solutions, 23, 87, https://doi.org/10.1007/s10291-019-0876-5, 2019.

Figure 1, Please avoid in legend such shortcuts as "With" and "Without". The Figure should be prepare in the way, which will make it possible for anyone to understand it content, without referring to the text. Response: Thank you for the suggestion. We revised Figure 1 as suggested.

Figure 4 has to be improved. The crossover time cannot overlap the crossover point, especially when green and red colors are used, because it makes it difficult to read them. The crossover time should not also overlap with the ship trajectory. Response: Thank you for the suggestion. We revised Figure 4 as suggested.

Please also note the supplement to this comment:
https://www.atmos-meas-tech-discuss.net/amt-2019-503/amt-2019-503-AC1-supplement.pdf

[Figure]

[Figure]

[Figure]

**Fig. 2.** Figure 4

**Supplement:**

---

## Referee Comment (RC2) · Anonymous Referee #3 · 2 Jun 2020

This research uses satellite, ground based, and ship borne PMV for validation comparison. The author tried to extract the clean points for satellite PMV validation but was not discussed in detailed how. The unclear scientific discussion makes the understanding this research frustrating. There are unfinished sentences and grammar mistakes. I can only recommend this manuscript accepted after a major revision. The scientific questions should be clarified, and the experiment design needs major improvement.

Major comments: 1. The author needs to read the guidelines for acronyms. Where should you use them and where should you define them in manuscript is a basic background knowledge for scientific writing. I stopped mentioning acronym problems at line 30, but much more corrections are needed in this manuscript about acronyms. 2. The result section is largely based on the separation of clean and contaminated pixels.

[Figure]

But I don't see a clear validation and explanation of how the author did this. 3. The separation from 200km, 150km, until 50km is nice, but the data points are too little. Overall 600+ points is quite small for this kind of comparison. Since you already have the automatically methods for running this analysis, I would highly suggest to extend the running to at least one full year. Minor Comments: 1. Line 12, please provide the full name of acronym PMV. 2. Line 14, I understand HY-2A is Haiyang-2A, but you need to add this to the end of Haiyang-2A for clarification. 3. Line 15, what is IGS stations? Acronym again. 4. Line 15, coastline along China or India? Before you submit a paper, it is always a good practice to get a second opinion. You are familiar with all the acronyms and station set ups, but not the readers. 5. Line 18, what is RMS? 6. Line 25, reference is needed here. 7. Line 25-27, the whole sentence needs to rewritten. Grammar mistake. 8. Line 30, Acronyms MODIS, FY-2C? SSM/I, TMI? If you will not mention these acronyms later simply write them in full name here. 9. Line 51, over the sea and over the land. 10. Line 71, delete the first and. The full sentence from line 71-72 needs to be rewritten. Grammar mistake. 11. Line 74, ECMWF is a large dataset, what exactly did you based on here? 12. Lin 123-127, more detailed coefficients calculation used here is needed. If you are using any standard lookup table, the reference should be provided. 13. Line 139-141, you named all the potential problem, what is you solution? Just say be careful is not enough. 14. Line 158-159, what is clean points? On both side how? How did you get the clear points? 15. Figure 2. Still what defines a clean point? Contaminated points? The better way is to first describe how the points are classified (more figures), then show a scatter plot of the PMV point's correction result. The current figure is very confusing. 16. Line 176-179, consider delete this paragraph, the sentences are useless and contain several gramma mistakes. 17. Line 184, the word complicated will raise concerns. Please elaborate on the advantage and disadvantages of the processing algorithm or packages. 18. Line 193, mislabeling Figure 3. Check the rest of figure labeling, most of them needs updates. 19. Line 203, how did you come up with the criteria of 200km and 2 hours, any histograms to show the overlapping points so that these criteria can be trusted?

---

## Author Comment (AC2) · 24 Jun 2020

Dear reviewer,

Thank you for the constructive and encouraging comments regarding our manuscript. We have enclosed a carefully revised manuscript according to the comments and suggestions provided. We also provide an item-by-item response to all comments.

Yours Sincerely, Zhilu Wu

Response to Reviewer

The author needs to read the guidelines for acronyms. Where should you use them and where should you define them in manuscript is a basic background knowledge

for scientific writing. Line 12, please provide the full name of acronym PWV. Line 14, I understand HY-2A is Haiyang-2A, but you need to add this to the end of Haiyang-2A for clarification. Line 18, what is RMS?Line 30, Acronyms MODIS, FY-2C? SSM/I, TMI?

Response: Thank you for the suggestion. We are so sorry about our mistake. We revised the manuscript accordingly: Line 12 is revised as "Ground-based Global Navigation Satellite Systems (GNSS) provide precise precipitable water vapor (PWV) with high temporal resolution". Line 10 is revised as "The calibration microwave radiometer (CMR) onboard Haiyang-2A (HY-2A) satellite". Line 15 is revised as "... using ground-based GNSS observations of 100 International GNSS Service (IGS) stations along ...". Line 18 is revised as "Geographically, the root-mean-square (RMS) is 1.12 mm in the polar region ...". Line 29 is revised as "satellite-borne infrared sensors (e.g. Moderate-resolution Imaging Spectroradiometer, Fengyun-2C) and microwave radiometers (Special Sensor Microwave/Image, Tropical Rainfall Measuring Mission's (TRMM) Microwave Imager (TMI))".

The result section is largely based on the separation of clean and contaminated pixels. But I don't see a clear validation and explanation of how the author did this.

Response: Thanks a lot for your comments. Before the validation, the raw HY-2A PWV observations were processed with a outlier detection method, and coastal observation reconstruction was implemented. The reconstruction method of contaminated HY-2A PWV is described from line 150 to line 160, and the HY-2A CMR observations (DOY 123 and 128, 2014) in Southeast Asia were used as an example to illustrate the coastal PWV reconstruction (line 161 to 175).

The separation from 200km, 150km, until 50km is nice, but the data points are too little. Overall 600+ points is quite small for this kind of comparison. Since you already have the automatically methods for running this analysis, I would highly suggest to extend the running to at least one full year.

Response: Thanks a lot for your comments. We acknowledge the number of crosspoints was limited in this study due to original GNSS observations. The limiting aspect of the study is the small number of points examined. And even fewer number of days which make the result not very much representative of general conditions. To better assess the HY-2A CMR bright temperature measurements and the PWV retrieving algorithms, more spatially distributed open-sea GNSS observations are needed, including those from various kinds of ships or buoys. Thank you so much for your advice again and we are planning to collect more GNSS observations (more than one year) to extends my research.

Line 15, coastline along China or India? Before you submit a paper, it is always a good practice to get a second opinion. You are familiar with all the acronyms and station set ups, but not the readers.

Response: Thank you for the suggestion. We revised the manuscript accordingly: "100 IGS stations along the global coastline "

Line 25, reference is needed here.

Response: Thanks a lot for your suggestion. We revised the manuscript accordingly: "Sea surface height measurement is mainly implemented by satellite altimetry, where the precise tropospheric delay is required to correct the atmosphere propagation error in the measured distance between satellite and sea surface (Obligis et al., 2011) "

Obligis E, Desportes C, Eymard L, et al. Tropospheric corrections for coastal altimetry[M]//Coastal altimetry. Springer, Berlin, Heidelberg, 2011: 147-176

Line 51, over the sea and over the land. Response: Thanks a lot for your comments. We revised the manuscript accordingly: "where it happens very often that one footprint covers partly over the sea and partly over the land "

Line 71, delete the first and. The full sentence from line 71-72 needs to be rewritten. Grammar mistake.

Response: Thank you so much for your advice again, we are truly sorry for the mistake.

СЗ

We revised the manuscript accordingly: "In this section, we introduced the processing strategy of ground-based and shipborne GNSS observations. The height correction for PWV of ground-based GNSS was also discussed."

Line 74, ECMWF is a large dataset, what exactly did you based on here?

Response: Thank you so much for the comment. We revised the manuscript accordingly: "Then we presented the HY-2A CMR retrieval method and the reconstruction algorithm of coastal HY-2A CMR contaminated data based on European Centre for Medium-Range Weather Forecasts (ECMWF) ERA-Interim layer data"

Line 123-127, more detailed coefficients calculation used here is needed. If you are using any standard lookup table, the reference should be provided.

Response: Thank you for the suggestion. The data we used is level 2 product, which is from National Ocean Satellite Application Center (NSOAS), Ministry of Natural Resources (MNR) of China, and our paper is focus on the validation of level 2 product (PWV data). We revised accordingly: "These coefficients must be estimated using external PWV datasets, e.g., NWM, radiosonde profiles, or previous satellite altimetry missions. In our study we used the product from National Ocean Satellite Application Center (NSOAS), Ministry of Natural Resources (MNR) of China."

Line 139-141, you named all the potential problem, what is you solution? Just say be careful is not enough.

Response: Thank you for the suggestion. We find the potential problem of CMR PWV data, therefore, we propose a method to reconstruct the contamination CMR data after line 143: "...Therefore, a linear fitting of HY-2A CMR PWV could be used for quality control in principle. ..."

Line 158-159, what is clean points? On both side how? How did you get the clear points? Figure 2 Still what defines a clean point? Contaminated points? The better way is to first describe how the points are classified (more figures), then show a scatter

plot of the PWV point's correction result. The current figure is very confusing.

Response: Thank you for the suggestion. The contaminated points are the CMR abnormal PWV data causing by the signal from land. The clean points are original HY-2A CMR PWV without contamination. We revised the manuscript accordingly: "In this study, the vertical integral of water vapor (VIWV) from ERA-Interim surface product was used, where the PWV differences between HY-2A CMR and ECMWF at crossover points should be small and stable. Those with extremely large values (differences over three times of the standard deviation value of the differences) were considered as contaminated points, and the remaining CMR data were taken as clean points."

Line 176-179, consider delete this paragraph, the sentences are useless and contain several gramma mistakes.

Response: Thank you for the suggestion. We deleted this paragraph.

Line 184, the word complicated will raise concerns. Please elaborate on the advantage and disadvantages of the processing algorithm or packages.

Response: Thank you for the comment. Before the validation, we need to pre-process raw CMR data. The CMR data flagged with "land" and "ice" were removed firstly, and then CMR data that the footprints within 100 km to GNSS sites were selected. The integral of water vapor (VIWV) from ERA-Interim surface product was used, and those with extremely large values (differences over three-time of the STD value of the differences) were considered as contaminated points. The pre-processing can help to find the coastal contaminated points and make preparation for the reconstruction, which make the validation result realiable.

Line 193, mislabeling Figure 3. Check the rest of figure labeling, most of them needs updates.

Response: Thank you for the suggestion. We revised the label of the Figure 3 and other figures in the manuscript.

Line 203, how did you come up with the criteria of 200 km and 2 hours, any histograms to show the overlapping points so that these criteria can be trusted?

Response: Thank you for the comment. Research shows that successive zenith wet delay estimates are significantly correlated for up to 1.7 hours (El-Mowafy A et.al., 2011). And a troposphere delay resolution of 1 or 2 hours is usually used in GNSS processing (Snajdrova K et . al., 2006; Geng J et. Al., 2012). Larger temporal resolution might miss the real signals, while too small temporal resolution might cause low robustness of the solution (especially for kinematic platforms). Wet delay is nearly proportional to the PWV. Therefore, we take 2 hours as the time criteria. There are only 49 crossovers when the distance criteria is 50 km, so we loose the distance criteria to 200 km to have more crossover points for comparison.

El-Mowafy A, Lo J. Prediction of troposphere wet delay [J]. Journal of Applied Geodesy, 2011, 5(3-4): 163-173. Geng J, Williams S D P, Teferle F N, et al. Detecting storm surge loading deformations around the southern North Sea using subdaily GPS[J]. Geophysical Journal International, 2012, 191(2): 569-578. Snajdrova K, Böhm J, Willis P, et al. Multi-technique comparison of tropospheric zenith delays derived during the CONT02 campaign [J]. Journal of Geodesy, 2006, 79(10-11): 613.